# Microwave quantum heterodyne sensing using a continuous concatenated dynamical decoupling protocol

Charlie J. Patrickson [1] ✉, Valentin Haemmerli [1], Shi Guo[1], Andrew J. Ramsay [2] & Isaac J. Luxmoore [1]

By sequentially recording the phase of an AC signal relative to an external clock, quantum heterodyne schemes have recorded MHz and GHz signals with Fourier-limited precision. However, in systems with large inhomogeneous broadening, existing heterodyne protocols provide limited protection of the spin coherence, impacting amplitude sensitivity. Here, we use a continuous microwave scheme that extends spin coherence towards the effective $T_2 \approx \frac{1}{2}T_1$ limit and resolves the frequency, amplitude and phase of MHz to GHz magnetic fields. In an ensemble of boron vacancies in hexagonal boron nitride the scheme achieves an amplitude sensitivity of $\eta \approx 3 - 5 \, \mu\text{T}/\sqrt{\text{Hz}}$ and phase sensitivity of $\eta_\phi \approx 0.076 \, \text{rads}/\sqrt{\text{Hz}}$. We demonstrate that the scheme is compatible with quantum heterodyne detection, recording a GHz signal with a resolution $< 1 \, \text{Hz}$ and SNR of 235 over a 10 s measurement. Achieving this performance in a two-dimensional material platform could have broad applications in probing nanoscale condensed matter systems.

Spin defects in wide-bandgap semiconductors are a promising platform for quantum sensing in ambient conditions and have been widely applied to the detection of magnetic fields at frequencies from DC to THz[1–4]. The frequency resolution of the sensor is limited by its interaction time with a signal field and therefore by the coherence time of the spin. The coherence time depends on the noise spectrum and the measurement protocol used to filter the noise. Dynamical decoupling schemes are particularly effective and can extend the coherence time to an effective limit of $T_2 \approx \frac{1}{2}T_1$[5]. Even so, frequency resolutions achieved with dynamical decoupling are typically limited, at room temperature, to hundreds of Hz for NV-center ensembles in diamond[3], tens of kHz for NV centres in silicon carbide[6], and hundreds of kHz to a few MHz for boron vacancies in hexagonal boron nitride[7–9]. To achieve higher resolution, quantum heterodyne measurement schemes have been employed. Here, the spin sequentially samples the phase evolution of a signal relative to an external clock, which has coherence many orders of magnitude better than $T_1$[10,11]. Whilst the frequency resolution is determined by the external clock, the bandwidth and sensitivity are determined by

the coherence time, and therefore dynamical decoupling sequences are generally used to sample the phase.

Dynamical decoupling is most commonly implemented using trains of pulses, where the sensor bandwidth is controlled by the interpulse spacing. This approach has been widely adopted in heterodyne sensing to detect signals at MHz frequencies and below[6,9–12], however, it can also extend the sensor response to the GHz range by dressing the spin resonance in a Mollow triplet[13]. Continuous schemes provide an alternative approach to pulsed dynamical decoupling. Spinlock sequences[14] use a strong microwave drive to match the Rabi frequency to the signal of interest and extend spin coherence towards $T_1$. These schemes are compatible with heterodyne readout, but are ultimately restricted to the MHz range by the power requirements and/or efficient delivery of microwaves to the spin. This limitation can be overcome with continuous concatenated dynamical decoupling (CCDD)[15,16], where the continuous microwave drive is phase or amplitude modulated, effectively screening the spin from low frequency noise. With this approach the same effective limit of $T_2 \approx \frac{1}{2}T_1$[3,17–19] achieved and control of the microwave drive provides tunability of a

[1]Department of Engineering, University of Exeter, Exeter, UK. [2]Hitachi Cambridge Laboratory, Hitachi Europe Ltd., Cambridge, UK. ✉e-mail: cp728@exeter.ac.uk

narrow sensing bandwidth that can extend from the MHz up to the tens of GHz range at room temperature[3,7]. Previous implementations have remained insensitive to signal phase to optimise sensitivity to signal amplitude and frequency[3,7], however this has precluded their use in quantum heterodyne protocols, which require the signal phase to be sampled sequentially.

In this work, we present a CCDD sensing protocol that can detect the phase, frequency and amplitude of an AC magnetic field, using coherence times approaching $T_2 \approx \frac{1}{2}T_1$. The scheme uses phase modulated CCDD to drive the spin along two different axes, at two different frequencies. A signal that is resonant with these rotations will cause the spin to deviate from the CCDD driven trajectory, with a path that depends on the signal phase. Information about the new trajectory, and therefore signal phase, is revealed by projecting the spin onto the z-axis through optical readout. We demonstrate the scheme using a $V_B^-$ ensemble in hexagonal boron nitride, where it successfully suppresses the effects of magnetic noise from the host III-V nuclear spin bath and leads to amplitude and phase sensitivity of $\eta \approx 3 - 5\,\mu T/\sqrt{Hz}$ and $\eta_\phi \approx 0.076\,rads/\sqrt{Hz}$, respectively. We show that the microwave drive parameters can be used to switch the spin response between phase and amplitude detection and use the CCDD phase sensitive sequence in a quantum heterodyne protocol, achieving a frequency resolution of 0.118 Hz at ~2.31 GHz, over a total measurement time of 10 s. The scheme provides a method of comprehensive characterisation of AC signals across a wide frequency range from MHz to the high frequency limit of the electron spin resonance (10s GHz at room temperature), and is equally applicable to a variety of solid-state defects and trapped atom and ion systems.

## Results

### Experimental setup

The device, shown in Fig. 1a is constructed of a sapphire substrate, patterned with a gold co-planar waveguide (CPW), which delivers the microwave drive fields to an ion-irradiated hBN flake on top. A 488 nm laser is modulated by an acousto-optic modulator, and focused through an NA = 0.55 microscope objective for optical excitation. The same objective collects photoluminescence, which is detected using a single photon avalanche detector (SPAD). The microwave drive and signal waveforms are generated using an arbitrary waveform generator (AWG). All results were recorded at room temperature in air (see methods for further experimental details).

The boron vacancy hosts two unpaired electrons, producing a spin-1 triplet ground state aligned along the principal crystal axis. A simplified representation of this system and the relevant optical states are presented in Fig. 1b. The $m_s = 0$ and $m_s = \pm 1$ spin states are separated by a zero field splitting (ZFS) of $D \approx 3.479$ GHz, and a strain field of $E \approx 59$ MHz[19]. We use off resonant optical pumping to prepare the spin into the $m_s = 0$ state. The same optical excitation provides readout of the spin state, where the emitted photoluminescence (PL) is brighter for the $m_s = 0$ than the $m_s = \pm 1$ states. A DC-magnetic field applied along the c-axis of the hBN, $B_z \approx 207$ mT, produces a Zeeman shift of $\omega_{\pm 1} = \pm \gamma_e B_z \approx \pm 5.74$ GHz. The frequency of the $m_s = 0 \Leftrightarrow m_s = -1$ transition used in these experiments is then $\omega_0 = |D - E - \omega_{-1}| = 2.32$ GHz.

The hBN is of natural isotopic composition, with approximately 99.6% $^{14}N$ nuclei of nuclear spin $I = 1$. The boron vacancy couples to the three nearest nitrogen nuclei[20], with a strong hyperfine interaction

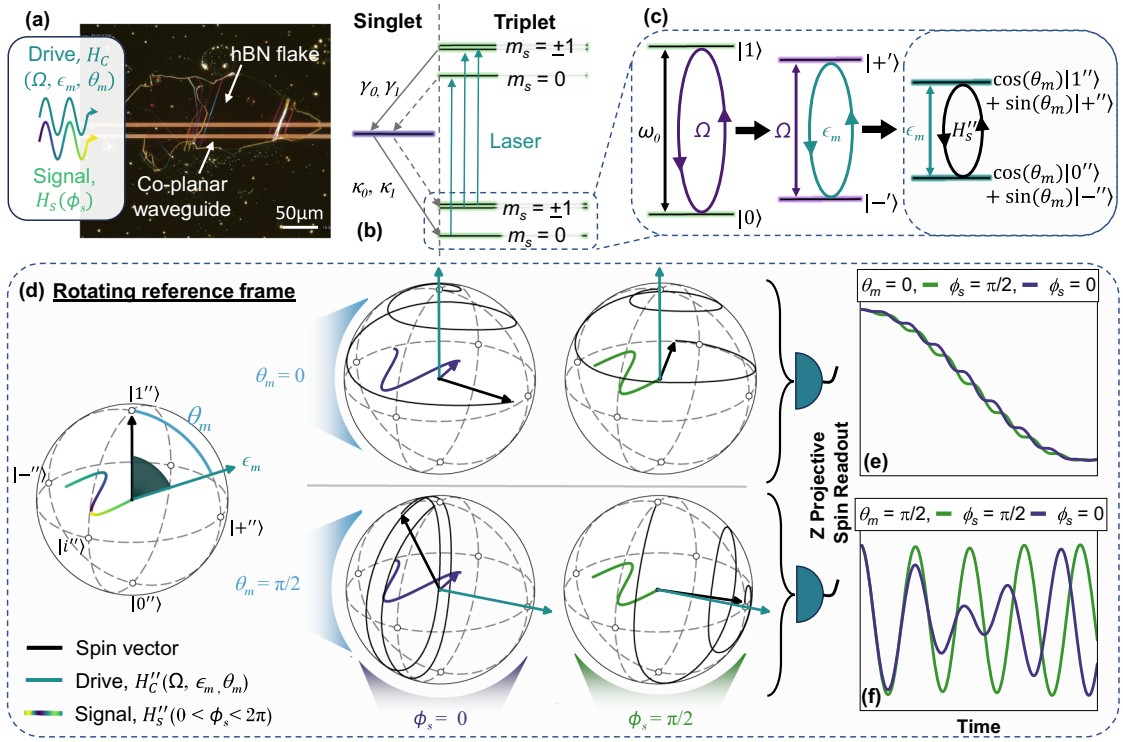

**Fig. 1 | Overview of the sample and protocol. a** Darkfield microscope image of the device. The CCDD microwave drive, $H_c(\Omega, \epsilon_m, \theta_m)$ and signal field $H_s(\phi_s)$ are applied to the $V_B^-$ spin ensemble via the co-planar waveguide. **b** Simplified model of the $V_B^-$ optical and spin states, with spin dependent decay rates, $\gamma_0$, $\gamma_1$, $\kappa_0$ and $\kappa_1$. **c** The CCDD drive field resonantly addresses the two-level system corresponding to the $|m_s = 0\rangle$ to $|m_s = -1\rangle$ spin transition, producing concatenated dressed states defined by the drive amplitudes $\Omega$ (purple) and $\epsilon_m$ (turquoise), and drive phase $\theta_m$. A resonant signal $H_s''$ will drive an additional transition between these states. ''s refer to the reference frame. **d** Signal driven $V_B^-$ spin evolution in the doubly rotating

reference frame. The frame is selected so that the drive field, $H_c(\Omega, \epsilon_m, \theta_m)$ (turquoise arrow), reduces to a DC component in the YZ plane, with the polar angle determined by the phase of the drive, $\theta_m$. A resonant signal field applied along the x-axis exerts a torque on the spin vector (black arrow). **e, f** The z-component of the subsequent spin trajectory is insensitive to signal phase if $\theta_m = 0$ (**e**), but is sensitive to phase if $\theta_m = \frac{\pi}{2}$ (**f**). This can be detected via optical readout, which projects the spin onto the z-axis. To demonstrate this the spin trajectory is plotted under two different signal phases, $\phi_s = 0$ (purple) and $\pi/2$ (green). Source data are provided via an open access repository (see Data Availability statement).

(HFI) of 47 MHz. Whilst strain induced variations of the zero-field splitting parameters[21–23] contribute to inhomogeneous broadening, fluctuations in the nuclear spin bath dominate, leading to short $T_2$ coherence times typically below 100 ns[19,20]. We have shown previously that CCDD can mitigate the effects of this inhomogeneous noise and extend the coherence time to a few microseconds[19] if the modulation amplitude is comparable to the linewidth.

## Phase detection using a double microwave drive

The central idea of CCDD is to use a pair of microwave drives to isolate the spin from magnetic noise. The first drive protects against low frequency phase noise. If applied in quadrature, the second drive counteracts fluctuations in the first drive[15]. When used in this way, a resonantly driven spin vector remains co-aligned with the second drive, creating a protected eigenstate. However, if exposed to a resonant signal field, effectively a third drive, the spin vector will deviate. The spin-projection along the z-axis is then detected optically and forms the basis of CCDD sensing schemes[3,7].

Whilst the signal phase influences the resulting spin trajectory, it has little impact on the z-projection (see Fig. 1e) and is therefore difficult to detect with optical readout techniques used in typical CCDD sensing schemes[3,7]. To sense the signal phase, we apply the second drive in phase with the first drive, perpendicular to the spin vector, so that it modulates the Rabi frequency. The signal driven spin trajectory is again dependent on signal phase, but crucially, so is the projection along the z-axis (Fig. 1f). This enables phase sensitive detection of microwave fields.

The dynamics are determined by the system Hamiltonian. Choosing to resonantly drive the $m_s = 0 \Leftrightarrow m_s = -1$ transition we approximate a two-level system, $H_0 = \frac{1}{2}\omega_0\sigma_z$. The CCDD drive field, $H_C$, acts along the x-axis and in general, the signal field $H_s$ can be applied along an arbitrary axis,

$$H = H_0 + H_C + H_s,$$
$$H_C = \Omega\cos(\omega_0 t - \frac{2\epsilon_m}{\Omega}\sin(\omega_m t - \theta_m))\sigma_x, \quad (1)$$
$$H_s = (g_x\sigma_x + g_y\sigma_y + g_z\sigma_z)\cos(\omega_s t + \phi_s),$$

where $\omega_0$ is the energy gap of the two level system, $\Omega$ and $\epsilon_m$ describe the amplitudes of the first and second CCDD drive fields, respectively. The dynamics of the optically detected spin z-component are sensitive to the signal amplitude $g_i$, the signal frequency $\omega_s$ and, if the drive phase is $\theta_m = \frac{\pi}{2}$, the signal phase, $\phi_s$.

To interpret how this complex Hamiltonian drives the spin vector we transform the system into a rotating reference frame. We consider a frame that tracks with the Larmor precession of the spin vector around the z-axis, defined by $\omega_0\sigma_z$, and have assumed that the drive field is resonant with the precession. Applying the transformation, $H' = e^{i\omega_0 t\sigma_z/2}He^{-i\omega_0 t\sigma_z/2} - \frac{1}{2}\omega_0\sigma_z$, we find,

$$H' = H'_C + H'_s,$$
$$H'_C = \frac{1}{2}\Omega\sigma'_x + \epsilon_m\sin(\omega_m t - \theta_m)\sigma'_y, \quad (2)$$
$$H'_s = \frac{g_x}{2}(\sigma'_x\cos((\omega_s - \omega_0)t + \phi_s) + \sigma'_y\sin((\omega_s - \omega_0)t + \phi_s)),$$

where primes denote the reference frame, we have applied the rotating wave approximation (RWA) and assumed $\epsilon_m \ll \Omega$. The transformation shifts all AC frequencies by $\omega_0$, such that the resonant drive $H_C$ reduces to a DC component applied along the x-axis, $\frac{1}{2}\Omega\sigma'_x$, and a second drive field, $\epsilon_m\sin(\omega_m t - \theta_m)\sigma'_y$.

The time dependence of the drive field $H'_C$ can be removed by transforming to a second reference frame, here with respect to the Rabi oscillation driven by $\frac{1}{2}\Omega\sigma'_x$. Applying $H'' = e^{i\Omega t\sigma'_x/2}H'e^{-i\Omega t\sigma'_x/2} - \Omega\sigma'_x/2$, for a resonant second drive where $\Omega = \omega_m$ we find,

$$H'' = H''_C + H''_s = \frac{\epsilon_m}{2}[\sin(\theta_m)\sigma''_y + \cos(\theta_m)\sigma''_z] + \frac{g_x}{2}(\sigma''_x\cos((\omega_s - \omega_0)t + \phi_s)), \quad (3)$$

where we have applied the RWA. The CCDD drive reduces to a time-independent magnetic field, $\epsilon_m$, which points in the $YZ''$ plane at a polar angle described by the phase of the drive, $\theta_m$. This is illustrated in Fig. 1d. A benefit of using this approach is that the CCDD dressed spin states produce six tuneable transitions centred on the electron spin resonance. This means that our device can select signal frequencies across a ~ 300 MHz range at a fixed DC field[7]. In Eq. (3) we omit five of the resonances and consider only the $\omega_s = \omega_0 - \epsilon_m$ resonance, as the sensor couples more strongly to the signal field here[7,24]. For off-resonant signals and remaining sensor resonances, see Supplementary Note 2 and Supplementary Note 3, respectively.

$H''$ sets the spin trajectory via the Heisenberg equation, $\dot{\sigma}'' = i[H'', \sigma'']$. To provide a more intuitive interpretation, we re-express this as the rotation $\dot{\sigma}'' = \mathbf{H}''(t) \times \sigma''$. To make the device sensitive to signal phase, we cast $\epsilon_m$ along the $y''$-axis by choosing $\theta_m = \frac{\pi}{2}$. This causes the spin vector to rotate in the $XZ''$ plane at the frequency $\epsilon_m$. Mixing the rotating spin with a signal oscillating at the frequency $\epsilon_m$ in the rotating frame will produce an angular velocity acting on the spin vector, causing it to rotate. Crucially, the coupling strength depends on the signal phase. To illustrate this interaction, in Fig. 1d we plot the trajectories of two different spin vectors exposed to signals of phase $\phi_s = 0$ (purple), and $\phi_s = \frac{\pi}{2}$ (green) (see methods). Evolution occurs in the same doubly rotating frame described above. The torque produced by the signal is largest for $\phi_s = \frac{\pi}{2}$, causing a rapid divergence of the spin vector from its original trajectory, which lay in the $XZ''$ plane. The coupling strength is minimised for $\phi_s = 0$. The cumulative impact on $\dot{\sigma}$ can be seen after $\epsilon_m t = 4\pi$, where the z-projection of the spin vector is drastically different for $\theta_m = 0$ in Fig. 1e and $\theta_m = \pi/2$ in Fig. 1f.

## Phase sensitive microwave detection

To probe the CCDD driven spin dynamics in the presence of a resonant signal, we perform Rabi experiments with the pulse sequence shown in Fig. 2b. The sequence samples the z-component of the spin trajectory after simultaneous application of control and signal microwave pulses of duration, $T_{MW}$ ($T_{MW} + \Delta T$), with optical readout measurements $P_T$ ($P_{T+\Delta T}$). These values are then used to calculate a normalised contrast, $C_{\Delta T} = (P_T - P_{T+\Delta T})/P_{T+\Delta T}$, where $\Delta T = 5\,\text{ns} = \frac{\pi}{\omega_m}$ to nullify the effects of $T_1$ decay. In Fig. 2a we use this sequence to illustrate the spin response when the drive phase $\theta_m = \frac{\pi}{2}$, for a signal amplitude of $g_x = 2$ MHz, frequency $\omega_s = \omega_0 - \epsilon_m = 2.31$ GHz, and signal phases of $\phi_s = 0$ (purple) and $\phi_s = \frac{\pi}{2}$ (green). The inset depicts the marked difference in spin response for the two phases, and can be visualised by the Fourier transforms in Fig. 2c where we find a pair of nested Mollow triplets. The central frequency is produced by the CCDD drive at $\Omega = 100$ MHz. The sidebands at $\Omega \pm \epsilon_m = 90$ and 110 MHz are produced by mixing with the second CCDD drive. The splitting of the central peak and sidebands when $\phi_s = \frac{\pi}{2}$ is caused by the signal field $g_x/2 = 1$ MHz[7]. These dynamics reflect the trajectories plotted in Fig. 1d, however they are superimposed onto the frequency components of the rotating reference frame, introducing additional complexity in the spin response. Note that the Fourier frequencies are independent of (dependent on) signal amplitude, $g_x$, for signal phases of $\phi_s = n\pi$ ($\phi_s = \frac{n\pi}{2}$), where $n$ is an integer (Fig. 2d and Supplementary Note 4). To highlight the distinction with CCDD sensing schemes that omit drive phase, in Fig. 2e we plot Rabi measurements with the drive phase $\theta_m = 0$. Here the plots are qualitatively the same, and the Fourier transforms in Fig. 2f, g confirm that the sensor response is independent of signal phase.

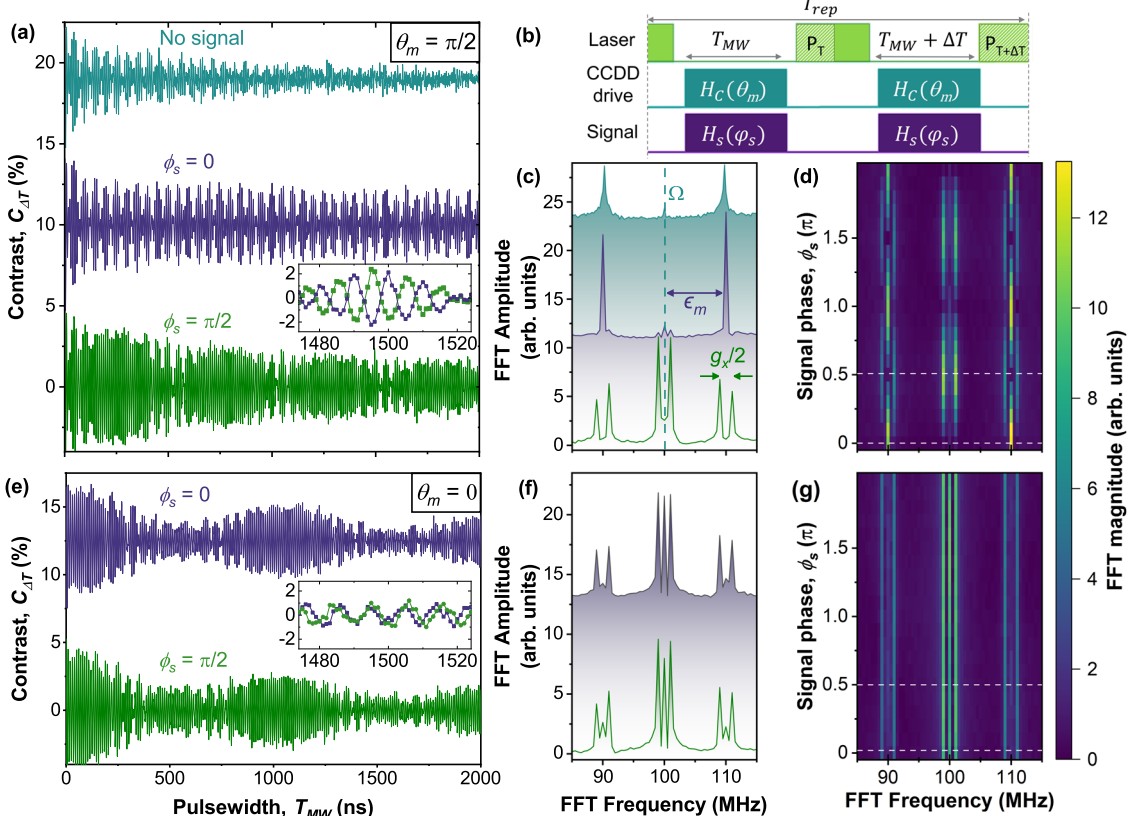

**Fig. 2 | $V_B^-$ spin dynamics driven by a CCDD field, $H_c(\theta_m)$, and a signal, $H_s(\phi_s)$.** The signal is resonant with the $\omega_s = \omega_0 - \epsilon_m = 2.31$ GHz sensor transition. **a** Phase-sensitive detection. Rabi measurements with the drive phase $\theta_m = \frac{\pi}{2}$, plotted without (turquoise) and with a signal applied for signal phases of $\phi_s = 0$ (purple) and $\phi_s = \frac{\pi}{2}$ (green). The inset highlights both the extended spin coherence in the presence of the signal, and the different sensor responses for the two signal phases. **b** Experimental sequence used to record the plots shown in (**a**) and (**c–g**). The contrast is recorded as $C_{\Delta T} = (P_T - P_{T+\Delta T})/P_{T+\Delta T}$, where the reference readout is measured with $\Delta T = 5$ ns $= \frac{\pi}{\omega_m}$ to cancel the effects of $T_1$ decay[29]. **c** Fourier transforms of (**a**). The signal enhances the FFT amplitude for signal phases of $\phi_s = 0$, and

creates additional Fourier components for $\phi_s = \frac{\pi}{2}$, where the spectrum is offset for clarity. **d** Equivalent of (**c**) for signal phases of $0 < \phi_s < 2\pi$, showing a smooth transition between two response regimes defined by $\phi_s = n\pi$ and $\phi_s = \frac{n\pi}{2}$. Dashed white lines correspond to spectra shown in (**c**). **e** Phase insensitive detection. Rabi measurement where the drive phase $\theta_m = 0$. The inset highlights similar sensor responses for the different signal phases. **f** Fourier transform of (**e**). **g** Fourier response for signal phases $0 < \phi_s < 2\pi$. Dashed white lines correspond to spectra shown in (**f**). Source data are provided via an open access repository (see Data Availability statement).

The phase and amplitude sensitivity are expected to scale with the coherence time $T_{CCDD}(\Omega, \theta_m, \epsilon_m, g_x)$[3]. We therefore quantify the expected sensing performance by measuring the spin coherence under different operating conditions. The upper limit is governed by $T_1 \approx 15\,\mu s$ in this sample, whilst the lower limit is determined by the Rabi time constant $T_{Rabi}(\Omega = 100\,\text{MHz}) = T_{CCDD}(\Omega = 100\,\text{MHz}, \epsilon_m = 0) \approx 36\,\text{ns}$ (see Supplementary Note 1 and Supplementary Fig. 1). By comparison, with no signal the CCDD drive achieves a coherence time of $T_{CCDD}(\Omega = 100\,\text{MHz}, \theta_m = \pi/2, \epsilon_m = 10\,\text{MHz}, g_x = 0) \approx 700\,\text{ns}$ (Fig. 2a) and can be further improved to a maximum of 1.2 $\mu s$ by optimizing $\epsilon_m$ (Supplementary Fig. 1e). When a signal is applied, the coherence time is found to improve, as the signal acts as an additional decoupling drive[3] and reaches $T_{CCDD}(\Omega = 100\,\text{MHz}, \theta_m = \pi/2, \epsilon_m = 10\,\text{MHz}, g_x = 400\,\text{kHz}) \approx 4\,\mu s$ (Supplementary Fig. 1f). This protection is not as effective as when $\theta_m = 0$, where we find $T_{CCDD}(\Omega = 100\,\text{MHz}, \theta_m = 0, \epsilon_m = 10\,\text{MHz}, g_x = 0) \approx 7.5\,\mu s$, but $\theta_m = \pi/2$ is necessary for phase sensitive detection, and improves the sensitivity by $\sim 30\times$ in the limit of low signal amplitude and $> 100\times$ for optimum signal amplitude.

To operate the device as a sensor we use the pulse sequence depicted in Fig. 3(c). The contrast $C_0 = (P_T - P_0)/P_0$ is recorded at a single pulsewidth of $T_{MW} = 950$ ns, which corresponds to a peak in the CCDD Rabi oscillation[25]. The reference readout $P_0$ is collected with the CCDD MW drive, $H_c(\theta_m) = 0$, and signal, $H_s(\phi_s) = 0$. Turning the CCDD drive off is useful in the case of an unknown continuous signal as the

dressed spin states are removed, providing a robust reference measurement that is unresponsive to the signal field. In Fig. 3a we record the contrast $C_0$ as a function of signal amplitude for signal phases of $\phi_s = 0, \frac{\pi}{2}$. The diverging responses demonstrates the sensitivity to signal amplitude and phase. In Fig. 3b we again compare to the case of $\theta_m = 0$, where the response is sensitive to signal amplitude, but not signal phase.

To benchmark the sensors performance, we calculate the amplitude sensitivity,

$$\eta = \frac{S(t_m)}{\max|\frac{\partial(\Delta C_0)}{\partial g_x}|}\sqrt{t_m},\qquad(4)$$

where $g_x$ is the signal amplitude, $\Delta C_0 = |(C_0(g_x) - C_0(g_x = 0)|$ describes the change in contrast due to the signal field, $S(t_m)$ is the standard deviation in $\Delta C_0$, and $t_m$ is the measurement time. To calculate $\max|\frac{\partial(\Delta C_0)}{\partial g}|$ we plot the change in contrast, $\Delta C_0$ as a function of signal amplitude, $g_x$, for different drive and signal phases in Fig. 3(c). Each data point is averaged over 10 measurements to provide an estimate of the standard deviation, $S(t_m)$.

For $\theta_m = \frac{\pi}{2}$, this gives sensitivities of $5.1\,\mu T/\sqrt{Hz}$ and $3.4\,\mu T/\sqrt{Hz}$ for signal phases of $\phi_s = 0$ and $\phi_s = \frac{\pi}{2}$, respectively. These values are comparable to other AC magnetometry schemes using $V_B^-$ ensembles[7,9], and as a reference for our device we find $\eta = 2.5\,\mu T/\sqrt{Hz}$

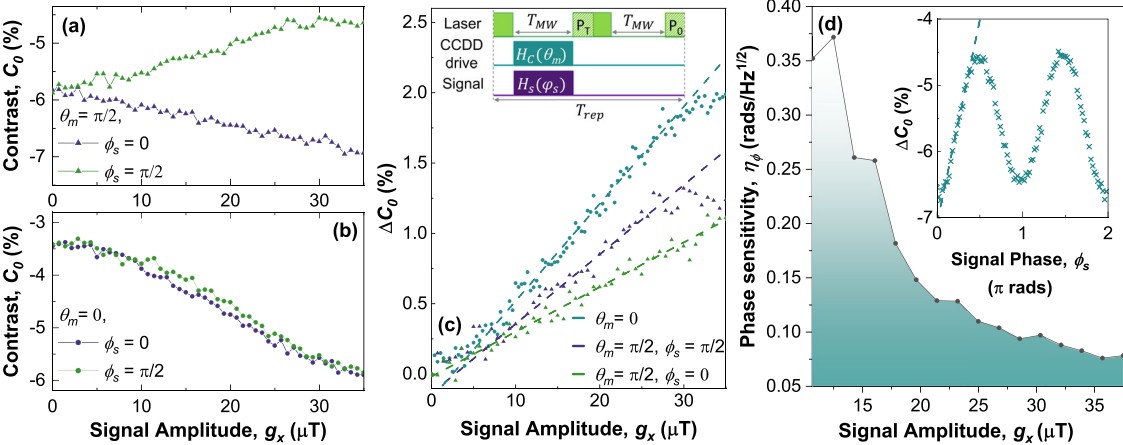

**Fig. 3 | Benchmarking of the sensor response to signal amplitude and phase.**
Contrast as a function of signal amplitude for $\theta_m = \frac{\pi}{2}$ in **a** and $\theta_m = 0$ in **b**, plotted for signal phases of $\phi_s = 0$ (purple), $\frac{\pi}{2}$ (green). The inset of (**c**) describes the experimental sequence used for (**a**–**d**), with a fixed pulsewidth of $T_{MW} = 950$ns, and the contrast calculated as $C_0 = (P_T - P_0)/P_0$. **c** The change in contrast with and without a signal applied as a function of signal amplitude. Linear fits provide $\max\left|\frac{\partial(\Delta C)}{\partial B}\right|$ for the sensitivity calculation in Eq. (4). **d** Phase sensitivity as a function of signal amplitude. A value for $\max\left|\frac{\partial(\Delta C)}{\partial \phi_s}\right|$ was required for each phase sensitivity calculation. The measurement used for a signal amplitude of 32 μT is provided in the inset as an example. Source data are provided via an open access repository (see Data Availability statement).

in the case of zero drive phase, $\theta_m = 0$. This suggests that the protocol is not only able to resolve signal phase, but does so with only a modest decrease in amplitude sensitivity.

The phase sensitivity $\eta_\phi$ is characterised as a function of signal amplitude $g_x$, and plotted in Fig. 3d. As before, the sensitivity is calculated using the contrast $\Delta C_0$ shown in the inset to Fig. 3(d). The sensitivity improves with signal amplitude due to a combination of the oscillatory response of the contrast, and a coherence time that improves with signal amplitude. A minimum in the phase sensitivity of $\eta_\phi = 0.076$ rads$/\sqrt{\text{Hz}}$ is reached for signal amplitudes of $g_x \approx 35 \mu$T. For a given signal amplitude the phase sensitivity can be optimised by selecting $T_{MW}$ to reduce the total measurement time, $t_m$ (see Supplementary Note 5). Note that the signal amplitude can also be measured independently of signal phase by choosing $\theta_m = 0$[7].

## CCDD heterodyne sensing

In the measurements of Fig. 3 the sequence repetition time, $T_{rep} = 5 \mu$s, and the phase measurement is averaged over $2 \times 10^5$ measurements (total measurement time, $t_m = 1$ s). The signal phase is set to be the same at the beginning of each measurement period, as would be the case for an interferometric measurement of a signal exactly resonant with $\omega_0 + \epsilon_m$. However, the phase sensitivity also allows this method to be adapted for quantum heterodyne sensing[10,11], where the instantaneous phase information is recorded for each individual measurement. Figure 4a provides an overview of the CCDD quantum heterodyne sensing technique. The CCDD drive field acts as the local clock and provides a stable coherent phase reference, and is switched on for a time $T_{MW}$ during each sensing sequence. The instantaneous phase difference between the continuously applied signal and the local clock provided by the CCDD drive is encoded in the PL intensity originating from the subsequent laser pulse. In this way, the relative phase is tracked as a function of time, thereby sampling the beat frequency between signal and clock. Taking the Fourier transform reveals the detuning of the signal from the clock, with a frequency resolution determined by the total number of measurements multiplied by $T_{rep}$.

Figure 4b shows the experimental realisation of the CCDD quantum heterodyne method, where an excerpt of a spin dependent PL time trace is plotted. To record the data, a continuous signal of frequency 2310.008 MHz was generated by the AWG and applied via the CPW. The $\omega_0 + \epsilon_m = 2310$ MHz resonance provided the $H_c$ clock frequency. An average of 1.8 photons were recorded per readout

sequence. The time sequence data is recorded for a total measurement time, $t_m = 10$ s. The data is autocorrelated, then fast Fourier transformed[13], as plotted in Fig. 4c. The dominant Fourier component $\omega_f$ corresponds to the signal demodulated by the clock at 16 kHz, and has a full-width half maximum of 0.118 Hz. Note that the peak appears at twice the true detuned frequency of $\delta = \omega_s - \omega_0 - \epsilon_m = 8$ kHz. This is because the sensor response cycles twice for signal phases between $0 \geq \phi_s \geq 2\pi$, as can be seen in the inset of Fig. 3d. Comparing the peak height to the standard deviation of the baseline gives a signal to noise ratio, SNR = 235. Notably, a single measurement can resolve multiple frequencies (see Supplementary Note 7 and Supplementary Fig. 9), demonstrating that CCDD could be applied to the detection of more complex signals that use carrier waves with frequency or amplitude modulation, for example.

For a fixed set of CCDD drive parameters, the device can detect detuned signals within a 100 kHz range of the sensor resonance. This is demonstrated in Fig. 4d, where we plot the peak frequency from the autocorrelation FFT as a function of signal detuning. This range is determined by the measurement repetition rate, $1/T_{rep} = 400$ kHz, which sets the Nyquist frequency of the PL time trace. This limit could be extended by reducing $T_{MW}$ from 950 ns to tens of ns (see Supplementary Note 5) or by shifting the sensor resonance with the $H_c$ drive amplitude, $\epsilon_m$[7]. Although 4c shows that the sign of the detuning is not implicit from a single measurement, this can also be deduced using $\epsilon_m$. Finally, in Fig. 4d we characterise how the SNR scales with measurement time, confirming that SNR $\propto t_m$ (green) with autocorrelation, compared to a shot noise limited SNR $\propto \sqrt{t_m}$ without (blue)[13]. The SNR of the heterodyne measurement is dependent on the signal amplitude (see Supplementary Note 6 and Supplementary Fig. 8) because it is underpinned by the phase sensitive CCDD protocol, the sensitivity of which depends on the signal amplitude (Fig. 3d).

## Discussion
To conclude, we have demonstrated a phase modulated CCDD sensing protocol that extends the coherence time of a $V_B^-$ ensemble into the μs regime by suppressing the effect of magnetic noise from the nuclear spin bath, whilst providing tuneable control over the phase, frequency and amplitude response of the sensor. For the phase sensitive detection mode, the coherence time is improved from $T_{Rabi} \approx 36$ns to $T_{CCDD} \approx 1.2 \mu$s with no signal, and to $T_{CCDD} \gtrsim 4 \mu$s $\approx 0.25 T_1$ for an optimum signal amplitude. Benchmarking our sensor, we have measured

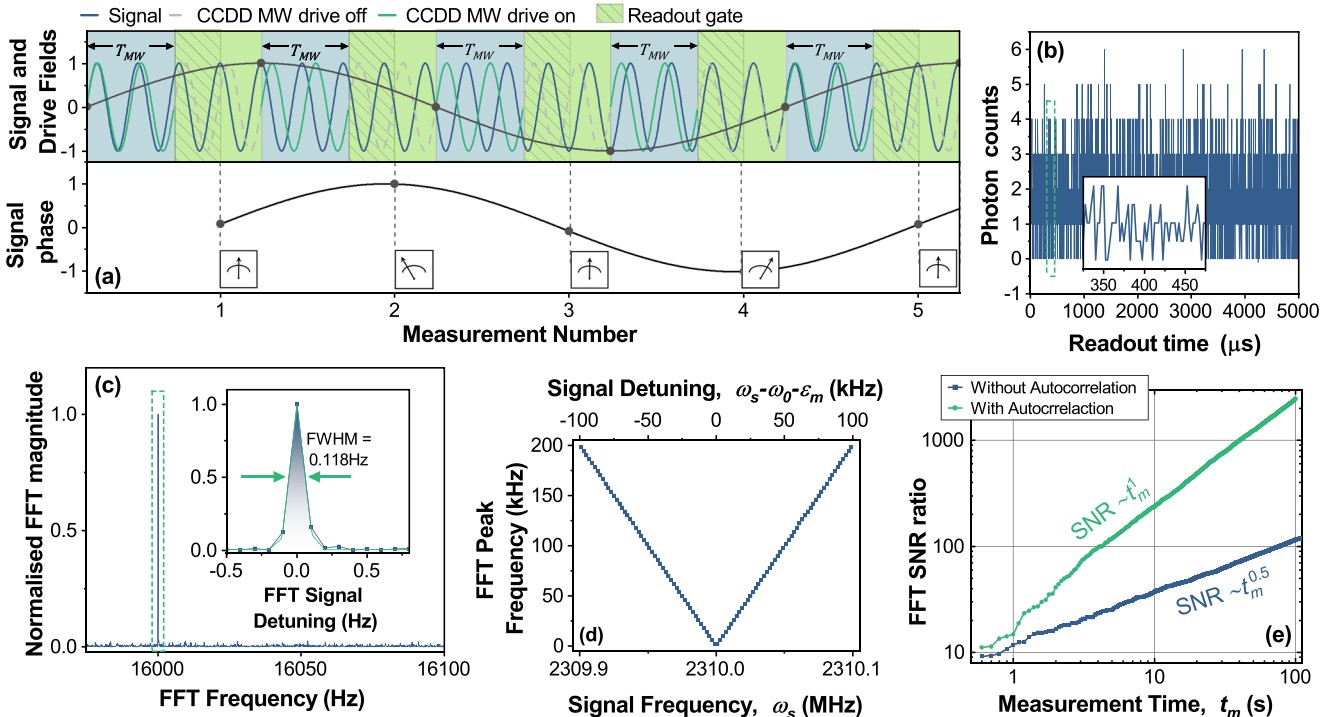

**Fig. 4 | Quantum heterodyne measurements using a continuous microwave drive. a** Schematic of the experimental sequence. A coherent microwave drive (turquoise) is used across sequential measurements, each lasting $T_{MW}$, to record the phase of a continuous, coherent signal (blue). The sensor output (bottom panel) records the relative detuning. **b** Experimental photon time trace, giving an excerpt of the sensor output analogous to the bottom panel of (**a**). The total measurement time $t_m$ was 10 seconds, with an average of 1.8 photons collected per measurement. The inset provides an example of the time trace in detail, from 325 to 475 $\mu$s. **c** FFT of the autocorrelation taken from (**b**). The inset shows a Gaussian fit providing a SNR of 235, with the FWHM giving a frequency resolution of 0.118 Hz. **d** Peak FFT frequency of the autocorrelated sensor output as a function of signal frequency. For the experimental parameters used here, the sensor can detect signals within a ± 100 kHz range of the CCDD sensor resonance. **e** Heterodyne SNR as a function of total measurement time, $t_m$. Taking the FFT of the autocorrelated data improves the SNR scaling from $\sqrt{t_m}$ to $t_m$. Source data are provided via an open access repository (see Data Availability statement).

amplitude and phase sensitivities of $\eta \approx 3 - 5\,\mu$T/$\sqrt{\text{Hz}}$ and $\eta_\phi \approx 0.076$ rads/$\sqrt{\text{Hz}}$, respectively. Finally, by embedding the sensing protocol in a quantum heterodyne measurement, we have measured a ~2.3 GHz signal with a resolution of ~ 0.1 Hz and SNR of 235, for total measurement time of 10 s. In previous work with $V_B^-$ ensembles in hBN[9], the coherently averaged synchronized readout (CASR) protocol[12] was used to detect an 18 MHz signal with resolution of 0.9 Hz and SNR of 8.5 for a 2 s sampling time. In both cases, the SNR is limited by a small sample volume. Note that while the results presented here focused on GHz signals, our protocol and experimental setup are also capable of detecting signals in the 10–150 MHz range[7]. Therefore, with room-temperature limited DC-magnetic fields ( < 1T) this scheme can be applied to the detection of magnetic fields in the range of 10 MHz to tens of GHz.

To compare our CCDD-based scheme with previous demonstrations of GHz heterodyne sensing[13,26], we consider the coherence time in the limit of low signal amplitude. In Staudenmaier et al.[26], this is $T_2^*$. In Meinel et al.[13], two schemes are presented: a continuous scheme, which is limited by $T_{Rabi} \gg T_2^*$, and a pulsed scheme, which is ultimately limited by $T_1$ but depends on the number of pulses used in the dynamical decoupling sequence. Pulsed dynamical decoupling schemes used in sensing[6,10–13] typically have between 8 and 32 $\pi$-pulses per readout, which limits the coherence time to ≲ 250 ns in $V_B^-$ ensembles[8,9]. Microsecond coherence times can be achieved with > 250 pulses[9], however sequences of this length typically require additional calibration procedures to correct cumulative pulse area errors. For example, a $\pi$-pulse fidelity of 99.9% would produce a net rotation of 45° in a 250 pulse sequence. It also becomes technically challenging to deliver sufficiently fast pulses for the readout time to be set to its

optimum value, i.e. equal to the coherence time[25]. We also note that long pulse sequences are accompanied by a large decrease in contrast[9]. In systems with large inhomogeneous broadening, each $\pi$-pulse is resonant with the central frequency of the ensemble. Detuned spins lose coherence with the drive, causing the contrast to decrease. Conversely, in CCDD schemes, the drive refocuses detuned spins for which the drive amplitude is larger than the detuning, allowing a large subensemble to be driven coherently[19]. Other continuous heterodyne schemes take advantage of the natural $\mu$s coherence times inherent to NV centres in isotopically purified diamond[18,26,27]. However, these are not an option with $V_B^-$ ensembles, where the natural spin state lifetime is < 100 ns[20,28]. In contrast, our scheme is able to extend $V_B^-$ coherence times into the $\mu$s regime, whilst remaining sensitive to signal phase, frequency and amplitude. It is important to acknowledge the significance of achieving this in a two-dimensional material, which can enable nanoscale sensor-source distances.

We note that the protocol supports lower frequency resonances at $\epsilon_m$ and $\Omega \pm \epsilon_m$[7], which could be relevant to nanoscale NMR. $\epsilon_m$ and $\Omega$ are responsible for coherence protection, and so the choice of spin system limits the frequency range of these resonances. Here we set $\epsilon_m = 10$ MHz, which we expect to be the minimum required to protect a $V_B^-$ ensemble in an hBN sample of natural composition[19]. For NV centres in diamond, the minimum can be set as low as $\epsilon_m \approx$ 100's kHz[3]. We expect these resonances to have similar amplitude and phase sensitivity to the GHz sensor resonances presented here, although we note that the $\epsilon_m$ resonance attenuates signal amplitudes by $g/4$[7], rather than $g/2$ (Fig. 2c). The high Rabi frequencies achieved in this work could enable nanoscale NMR at large magnetic fields > 1 T, accessing NMR frequencies in the 10's MHz, in comparison to

existing pulsed schemes which are limited to NMR frequencies below a few MHz[10-12].

The system presented here could also be applied as an effective probe of other low-dimensional condensed matter systems that present AC magnetic fields[29,30]. For instance, the out-of-plane DC field used in this work would support forward volume spin wave modes in ferromagnetic thin films[31,32]. These spin waves are a promising platform for next generation data transfer[31], as they avoid ohmic heat loss and support GHz to THz fields. For example, spin waves resonant with our device could be optically excited[33,34] at one end of the CPW. As these spin waves propagate with ≤100 μm wavelengths[35], the phase dependent sensor response could be used with time correlated laser scanning confocal microscopy to image the spin waveform along the length of the CPW, potentially providing new insight into spin wave dispersion relations[33,36]. Operating above saturation magnetisation would protect the defect ensemble from spin mixing produced by ferromagnetic in-plane fields. Although thermal and incoherently driven magnon modes are known to degrade $T_1$ times, which would limit the sensitivity of the scheme, this can be partially offset by displacing the ensemble by a small distance from the ferromagnetic surface[37]. Additionally, any associated increase in inhomogeneous noise can be decoupled with the CCDD drive, provided that the modulation amplitude is larger than the inhomogeneous linewidth. Finally, we note that this sensing scheme could also aid in the development of microwave circuitry to probe failure modes or ohmic heating in regions of high current loads[38-40]. More broadly, combining CCDD schemes with bright single spin defects[41], and using local nuclear spins as ancilla qubits, presents a promising platform for future quantum sensing endeavours.

## Methods

### Experimental

PL is excited using a 488 nm diode laser (Roithner SHD4850MG) pulsed by an acousto-optic modulator (AA Optoelectronics MT110-A1-VIS). An objective lens (Nikon LU Plan Fluor 50x BD, NA = 0.55) is used to focus the laser to a diffraction-limited spot on the hBN flake and at the center of the co-planar waveguide. Photoluminescence from the boron vacancy ensemble is collected with the same objective, filtered by a 750 nm long pass filter (Thorlabs FELH0750) and recorded on a single photon avalanche diode (SPAD, Excelitas SPCM-AQRH-15-FC). The microwave control and signal waveforms are generated using an arbitrary waveform generator (AWG, Keysight M8195A), amplified (Agilent 83017a) and applied via a circulator (Pasternak PE8432) to the CPW. The other end is terminated with 50 ohms. The optical and microwave excitation are synchronised with a digital pattern generator (Swabian PulseStreamer) and the photon counts recorded with time-tagging electronics (Swabian Time Tagger 20). All experiments applied a constant phase offset of 0.07 $\pi$ to the drive field, as this produced the optimum sensor response to signal phase in the device.

The maximum waveform length is limited by the AWG memory to approximately 2.5 ms. For simplicity, we choose a waveform length of 1 ms and ensure the frequency of all control and signal components is an integer multiple of 1 kHz. The 1 ms sequence is repeated resulting in signals with greater coherence than dictated by the memory constraints[13]. The optical and microwave control pulses are applied to the sample, along with the constant signal field for a total measurement time, $t_m$. All detected photons are time-tagged and saved. In post-processing a time gate is applied to keep only photons that arrive within a time $t_{gate}$ = 350 ns of the beginning of each optical pulse (repetition rate of 400 kHz). These photons are then binned according to the time that has elapsed since the beginning of the measurement, with an example shown in Fig. 4b. We fast Fourier transform the autocorrelation of this data to generate a spectrum of the signal, down-converted by the frequency of the double-dressed resonance, $\omega_0 - \epsilon_m$.

### Model

The Bloch-vector dynamics plotted in Fig. 1d were numerically calculated using the Heisenberg equation $\dot{\sigma'} = i[H'_C, \sigma']$, which is re-expressed as a rotation, $\dot{\sigma} = \mathbf{H}''(t) \times \sigma$, where $\dot{\sigma}$ is the time derivative of the spin vector, $\mathbf{H}''(t)$ is the interaction picture Hamiltonian as defined in the main text, and $\sigma$ is the spin vector. For each time step used in the model, a single effective field is calculated by first summing the vector components in $\mathbf{H}''(t)$. This forms an axis rotation for the spin vector, with the angle of rotation determined by the magnitude of the summed vector components. The process is repeated iteratively for all time steps to produce the plotted trajectories.

## Data availability

The data used in this study are available in the Open Research Exeter database under accession code https://doi.org/10.24378/exe.5687.

## Code availability

The underlying code for this study is available on request from the corresponding author.

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

## Acknowledgements

This work was supported by the Engineering and Physical Sciences Research Council [Grant numbers EP/S001557/1 and EP/L015331/1] (I.J.L.), Partnership Resource Funding from the Quantum Computing and Simulation Hub [EP/T001062/1](I.J.L.) and an Engineering and Physical Sciences Research Council iCASE in partnership with Oxford Instruments Plasma Technology (I.J.L. and C.J.P.). Ion implantation was performed by Keith Heasman and Julian Fletcher at the University of Surrey Ion Beam Centre. For the purpose of open access, the author has applied a 'Creative Commons Attribution (CC BY) licence to any Author Accepted Manuscript version arising'.

## Author contributions

I.J.L. and A.J.R. conceived and designed the experiments. V.H. and S.G. fabricated the sample. I.J.L and C.J.P. built the experimental setup and performed the measurements. I.J.L. and A.J.R. supervised the project. I.J.L., C.J.P., and A.J.R. analysed and discussed the experimental results. C.J.P and A.J.R. performed calculations. C.J.P. wrote the manuscript with contributions from I.J.L and A.J.R.

## Competing interests

The authors declare no competing interests.
