## [Transparent Peer Review file · Nature Communications]

Microwave quantum heterodyne sensing using a continuous concatenated dynamical decoupling protocol

Corresponding Author: Mr Charlie Patrickson

Version 0:

Reviewer comments:

Reviewer #1

(Remarks to the Author)

The manuscript by Patrickson et al. presents a modification of existing continuous heterodyne sensing schemes, making this protocol compatible with spin centers with short coherence times. The authors demonstrate the application of this protocol using the boron vacancy in hexagonal boron nitride (hBN) as a test system. This defect is one of the best-studied spin centers in 2D materials but faces practical limitations as a quantum sensor due to fast decoherence caused by interactions with surrounding nuclear spins. Since all naturally occurring isotopes of boron and nitrogen in hBN have non-zero nuclear spins, the total percentage of spinful isotopes in natural hBN is effectively 100%, resulting in significant magnetic noise. Nevertheless, sensing experiments with the boron vacancy can potentially unravel new fundamental physics of 2D materials.

The optimization of experimental protocols to mitigate fast decoherence is an important research direction with potential applications beyond the boron vacancy, and the authors present a novel method to achieve this goal. While the technical details of the instrumental part of the work are beyond the scope of my expertise, the physical model used to design and implement the proposed protocol appears to be sound. Therefore, I think the manuscript is suitable for publication in Nature Communications once the authors address the following suggestions/questions:

1. As described in lines 98-99, the rhombic anisotropy of the zero-field splitting (the E value) serves as one of the model parameters that define the transition frequency, which is further used to determine the frequency of the signal. However, a non-zero E reported in this work is likely caused by strain in the sample rather than being an inherent property of the boron vacancy itself (see <https://doi.org/10.1038/s41524-020-0305-x>). Since the strain is not necessarily uniform, the boron vacancies throughout the sample can feature varying transition frequencies. It might be helpful to add a short comment on whether the possible non-uniformity of the ZFS throughout the sample could affect the accuracy and efficiency of the proposed sensing protocol.
2. In the proposed sensing scheme, the applied signal field acts as an additional decoupling drive, extending spin coherence. As shown in the manuscript, this works well for an AC signal with a well-defined frequency, phase, and amplitude. The authors suggest that the protocol can also be used for detecting AC magnetic fields in real-life condensed matter systems. However, could the authors provide more specifics on the expected robustness of the sensing protocol to various sources of environmental noise and signal complexity?
3. There are two Methods sections—one on page 4 and another on page 16. Please consolidate these sections for clarity and coherence.

Reviewer #2

(Remarks to the Author)

The manuscript by Patrickson et. al extends their previous work (npj Quantum Information 10, 5 (2024)), which demonstrated high-frequency (up to GHz) RF/MW field sensing using VB- defects in hBN and a continuous concatenated dynamic decoupling (CCDD) scheme.

The additional feature shown here involves the sensitivity of the protocol to the signal phase, leading to the implementation of a heterodyne-like scheme for RF/MW detection with high spectral resolution. The manuscript is interesting, showcasing high-quality experiments and data.

However, I found some critical points that I would like to highlight:

Major:

- I am quite confused about how the claimed coherence protection contributes to the sensitivity of the method. For example, on line 59: "... we present a continuous drive sensing protocol that can detect the phase, frequency and amplitude of an AC magnetic field, whilst also achieving coherence times approaching $T_2 \approx 1/2 T_1$ ". However, they also mention that to achieve phase sensitivity, they have to compromise (at least partially) on coherence protection. Line 118: "To sense the signal phase, we apply the second drive in phase with the first drive, perpendicular to the spin vector, This reduces the protection provided by the microwave drive [30]". The authors should clarify this point, as it seems to contradict their earlier claims. Is coherence protection crucial for detection or not? To what extent? Additionally, what are the relaxation time T_1 and coherence times T_2 of the system in their experiments? What is the real meaning of the time constant T_{Rabi} given in the supplementary information, and is this time constant relevant for the CCDD experiment? How?

- Line 121: "Instead, coherence stabilization is now provided by the signal field (Fig. 2(a))." It would be appropriate to cite the work of Stark et al. (Nature Communications 1105, 2017), as they addressed a similar concept. The limits of this approach remain unclear to me. Specifically, how strong does the signal field need to be to function as an auxiliary protective field? Can this method still be effective in realistic sensing scenarios with very weak AC fields?

- Fig. 1 and description of spin dynamics: I acknowledge the authors' efforts to describe the spin dynamics as thoroughly as possible. However, to fully understand the experiments and the physics involved, I had to refer back to the literature, particularly the previous npj Quantum Information article by the same authors. The manuscript does not seem self-sufficient in this respect. At a minimum, the basic principles should be included so that readers can grasp the physics underlying the experiment.

- While I appreciate the organization of Figure 1, the authors should label the axes in the Bloch spheres shown in Figure 1(d). It would also be helpful to divide Figure 1(d) into labeled sub-figures, and refer to these sub-figures accordingly throughout the main text.

- Line 56: "Alternatively, single continuous drives have been used to replicate the phase response, [22, 26], but the sensitivity of these schemes is limited by coherence times $\ll T_2$ ".

Again, on Line 289: "A continuous scheme is also presented in [22], however, the single drive does not extend the coherence time, limiting the sensitivity."

Are the authors sure about this? Aren't these methods rather limited by $T_1\rho$, which is typically much longer than T_2 ? Isn't that an extension of the coherence? I would suggest double checking this point.

- Line 280 - In my opinion this statement needs to be double-checked: "The well-established CASR protocol[19] was recently demonstrated using a VB ensemble in hBN for the first time[38], where the authors achieved a frequency resolution of 0.9 Hz for an 18 MHz signal sampled for 2000 s, compared with 0.118 Hz resolution for a 10 s integration in this work." It's easy to verify that the authors of [38] sampled their signal for 2 seconds, not 2000 seconds (as shown in one of the main figures in their paper, Rizzato et al., Nat. Commun. 14, 5089 (2023)).

Minor:

Supplementary Note 1 was never mentioned throughout the text.

In the Supplementary Information, the title of the manuscript is missing, as well as the headings of Supplementary Notes 1, 2, and 3, etc...

Methods: It is good practice to state not only what kind of instrumentation was used, but also what specific equipment was used, so that one can, in principle, try to reproduce the experiments using the same machines.

Line 330: Double parentheses in reference 22.

Reviewer #3

(Remarks to the Author)

In their manuscript, Patrickson et al. report on a protocol for the detection of microwave (GHz) signals using defect spin qubits (ensemble of boron vacancies in hBN) as quantum sensors. The protocol is based on a continuous concatenated dynamical decoupling (CCDD). In addition to sensing the microwave amplitude, by selecting a proper phase of the second drive, the sensitivity to the signal phase can be attained. The protocol can also be combined with the quantum heterodyne scheme, and in this case provides a frequency resolution of 0.118 Hz in a 10-sec measurement.

The detection of both amplitudes and phases of high-frequency signals with high sensitivities is of importance in expanding the application range of quantum sensing. The manuscript is overall well-written, and the experimental data is rich and high quality. What bothers me however is the way in which the introduction is written. In my view, it is not very readable as well as failing to position the present work in a proper framework. So, the revision is necessary to reach my acceptance/rejection

decision. See my comments below.

In the first paragraph, the authors begin with the choice of the materials, listing up dense NV centers in bulk, NV centers in nanodiamonds, B vacancies and "single carbon-related defects of ambiguous structure" in hBN. These are quite remote from the main topic of the present work, CCDD-based high-frequency sensing. I understand that the coherence of a quantum sensor is a limiting factor to achieve high sensitivities, and the CCDD-based protocol is advantageous when the intrinsic coherence time of the sensor is short, such as in the case of B-vacancy in hBN used in the present work. But this is a bonus; what is more important is that the protocol itself is independent of the choice of the materials (so has wide applicability). I do not see a good reason to start the introduction this way.

The second paragraph outlines the existing sensing protocols, but it looks to me that the authors are trying to unduly undermine them. For instance, quoting Refs. [23] and [24], the authors state "pulsed methods are sub-optimal, as the sensitivity suffers from pulse-area errors." It is true that one should pay careful attention to the consequence of finite pulse lengths, but the issue raised in Ref. [24] can be trivially solved by including the pulse lengths in the total sensing time (which is almost always done). To highlight the effect, the examples in Ref. [24] is taken from the cases when the pulse durations are very long (~ 1 us), not the condition where many practical sensing experiments are carried out. The detuning effect discussed in Ref. [23] may be potentially more serious, but the effect is pronounced when the total number of pulses is large and XY8 is shown at least to suppress the spectral splitting. On the other hand, the quantum heterodyne protocol utilizes a short pulse sequence repeated with a constant interval. I am so far not aware of the cases when the quantum heterodyne protocols have practically suffered the effect of finite pulse lengths. If known, please cite appropriate references. Also, in this paragraph, the authors mention that the CCDD-based protocol achieves the effective coherence limit of $T_2 \sim (1/2)T_1$. Please provide (not necessarily in the introduction) the concrete values of T_1 and T_2 demonstrated by the current CCDD-based sensing experiments.

In the third paragraph, the authors introduce their protocol, starting with "In this work, ..." However, the key idea comes from CCDD, which has been extensively worked out in the past, in particular in Ref. [27]. Of course, Ref. [27] is cited, but why wasn't CCDD discussed before "In this work"? The introduction must clearly separate the past achievements from the present work. In fact, I feel at the conceptual level the novelty of the present protocol is not so high. In addition to Ref. [27], CCDD-based sensing in the GHz range has been demonstrated in Ref. [29]. In comparison to Ref. [29], the authors pitch that their protocol can sense the microwave phase as well. But this requires the signal being highly coherent. Rather, Ref. [29] points out that CCDD-based sensing can be done even when the driving and signal fields are not phase-matched so that the sensing is generic and widely applicable. These different viewpoints should be more clearly stated. The present work also has a significant overlap with the authors' own work of Ref. [34]. It is heavily cited in the technical part, but not in the introduction. Again, the difference between Ref. [34] and the present work should be clarified in the introduction. What is the conceptual advance, other than moving to the higher frequencies?

Minor errors I have noticed are listed below.

(1) There are multiple comments like "see Supplementary Note #" in the main text, but the sections of Supplementary Information (SI) are not numbered. Please add the section numbers.

(2) After the equations, "where" should start without spaces. (And in this case a comma should be added after the equations.)

(3) In some situations, the units for the amplitudes and phase sensitivities are given as " $\mu\text{T} \sqrt{\text{Hz}}$ " and " $\text{rads} \sqrt{\text{Hz}}$," respectively. Please check and correct them.

(4) In Line 57, "... the phase response, [22, 26], but..." should read "... the phase response [22, 26], but..."

(5) In Line 83, "A 488nm laser" should read "A 488-nm laser."

(6) In Line 84, "NA=0.55" should read "NA = 0.55."

(7) In Line 88, "room temperature in air. (see...)" should read "room temperature in air (see...)"

(8) In Line 99, ω_0 as calculated from the given D, E, ω_{-1} does not give 2.32 GHz. Note that ω_{-1} is given as negative.

(9) In Line 220, " $\eta_{\phi} = 0.076 \text{ rads} \sqrt{\text{Hz}}$ " should read " $\eta_{\phi} = 0.076 \text{ rads} \sqrt{\text{Hz}}$."

(10) In Line 254, "SNR=235" should read "SNR = 235."

(11) In Line 260, "950 ns to 10s of ns" should read "950 ns to tens of ns," to avoid confusion.

(12) In Line 267, the authors write "we have demonstrated a phase modulated CCDD sensing protocol...". As far as I can see, the authors controlled the drive phase θ_m , but this is hardly a phase modulation, which would usually mean the phase being time-dependent.

(13) In Line 315, "a 488 nm diode laser" should read "a 488-nm diode laser."

(14) In Line 316, "N.A.=0.55" should read "N.A. = 0.55." Also, the use of both "N.A." and "NA" (Line 84) should be avoided.

(15) In Line 330, "constraints [[22]]" should read "constraints [22]."

(16) In Line 333, " $t_{\text{gate}} = 350\text{ns}$ " should read " $t_{\text{gate}} = 350 \text{ ns}$."

(17) In the caption of Fig. 1 of SI, it will be better to include the coefficient in front of $\sin(\omega t)$. " $T_{\text{Rabi}} = 36\text{ns}$ " should read " $T_{\text{Rabi}} = 36 \text{ ns}$."

(18) In the caption of Fig. 3 of SI, " $T_{\text{M}W}$ " should read " T_{MW} ."

(19) In Line 21 of SI, "This illustrates our protocols dependence on..." may mean "This illustrates our protocol's dependence on..." See also Line 75 of SI.

(20) In Line 50 of SI, "it's direction of propagation" should read "its direction of propagation."

(21) In Line 113 of SI, "125ns" should read "125 ns." (And the unit should not be italicized.)

Version 1:

Reviewer comments:

Reviewer #1

(Remarks to the Author)

After reviewing the revised manuscript, I confirm that all the points I raised in the previous round have been addressed. The authors have clarified the issues and adjusted the manuscript accordingly. I reiterate my recommendation for publication in Nature Communications.

Reviewer #2

(Remarks to the Author)

While I find the manuscript somewhat incremental compared to the authors' previous works, the results presented are solid overall. In this revised version, the authors have addressed my suggestions thoroughly. Combined with the integrated feedback from other reviewers, these changes make the manuscript substantially more comprehensive and complete, while appreciably enhancing its clarity and strengthening the overall conclusions.

I do have one remaining request. Throughout the manuscript, the authors have extensively demonstrated the method's performance for sensing signals around 2 GHz. Moreover, in the conclusion, they primarily highlight the detection of spin waves in 2D materials as a specific application—one that benefits from sensing in the GHz range. However, since the manuscript (beginning with the title) appears to have shifted toward a more general development of a sensitive and versatile sensing protocol, I would appreciate further comments on how the technique performs in lower frequency regimes (e.g., 10–150 MHz). While the authors note that their protocol is capable of detecting signals in this window, they do not elaborate on its actual performance there. This frequency range would be highly relevant for another important application, such as magnetic resonance sensing. Such a discussion would be of considerable interest to a large segment of the quantum sensing community.

After addressing this point, I recommend acceptance of the manuscript.

Reviewer #3

(Remarks to the Author)

I am satisfied with the revisions made by the authors. The revised introduction provides a concise summary of the existing quantum sensing protocols whilst the advantage of the present protocol is more clearly described. Supplementary Note 1 and Supplementary Fig. 1 is also a concise summary of coherence protection provided by this protocol. In my view, a separation of these two (related but different) aspects made the manuscript more readable. I recommend the present manuscript for publication in Nature Communications.

Response to Reviewer Comments

We thank all three reviewers for their thorough reviews on our manuscript, for their positive comments and constructive criticism. In the following we provide a point-by-point response, detailing where we have amended the manuscript and added new data and analysis.

Reviewer #1 (Remarks to the Author):

The manuscript by Patrickson et al. presents a modification of existing continuous heterodyne sensing schemes, making this protocol compatible with spin centers with short coherence times. The authors demonstrate the application of this protocol using the boron vacancy in hexagonal boron nitride (hBN) as a test system. This defect is one of the best-studied spin centers in 2D materials but faces practical limitations as a quantum sensor due to fast decoherence caused by interactions with surrounding nuclear spins. Since all naturally occurring isotopes of boron and nitrogen in hBN have non-zero nuclear spins, the total percentage of spinful isotopes in natural hBN is effectively 100%, resulting in significant magnetic noise. Nevertheless, sensing experiments with the boron vacancy can potentially unravel new fundamental physics of 2D materials.

The optimization of experimental protocols to mitigate fast decoherence is an important research direction with potential applications beyond the boron vacancy, and the authors present a novel method to achieve this goal. While the technical details of the instrumental part of the work are beyond the scope of my expertise, the physical model used to design and implement the proposed protocol appears to be sound. Therefore, I think the manuscript is suitable for publication in Nature Communications once the authors address the following suggestions/questions:

1. As described in lines 98-99, the rhombic anisotropy of the zero-field splitting (the E value) serves as one of the model parameters that define the transition frequency, which is further used to determine the frequency of the signal. However, a non-zero E reported in this work is likely caused by strain in the sample rather than being an inherent property of the boron vacancy itself (see <https://doi.org/10.1038/s41524-020-0305-x>). Since the strain is not necessarily uniform, the boron vacancies throughout the sample can feature varying transition frequencies. It might be helpful to add a short comment on whether the possible non-uniformity of the ZFS throughout the sample could affect the accuracy and efficiency of the proposed sensing protocol.

(R1.1) Spatially varying zero-field splitting parameters contribute to the inhomogeneous broadening of the boron vacancy spin ensemble. However, continuous concatenated dynamical decoupling (CCDD) is effective in suppressing the inhomogeneous broadening when the modulation amplitude is comparable to the linewidth, as it synchronises spins detuned from the central drive frequency. This is thanks to the large Rabi frequency that can be achieved with the coplanar waveguide sample design. We have included extra references to note the contribution of strain on inhomogeneous broadening and emphasised that CCDD is effective in reducing its impact. The relevant paragraph can be found on P. 5, L. 103 and reads: “The hBN is of natural isotopic composition, with approximately 99.6% ^{14}N nuclei of nuclear spin $I = 1$. The boron vacancy couples to the three nearest nitrogen nuclei [19], with a strong hyperfine interaction (HFI) of 47 MHz. Whilst strain induced variations of the zero-field splitting parameters [20-22] contribute to inhomogeneous broadening, fluctuations in the nuclear spin bath dominate, leading to short T_2 coherence times, typically below 100 ns [18, 19]. We have shown previously that CCDD can mitigate the effects of this inhomogeneous noise and extend the coherence time to a few microseconds [18], if the modulation amplitude is comparable to the linewidth [23].”

2. In the proposed sensing scheme, the applied signal field acts as an additional decoupling drive, extending spin coherence. As shown in the manuscript, this works well for an AC signal with a well-defined frequency, phase, and amplitude. The authors suggest that the protocol can also be used for detecting AC magnetic fields in real-life condensed matter systems. However, could the authors provide more specifics on the expected robustness of the sensing protocol to various sources of environmental noise and signal complexity?

(R1.2) We have now tested the protocol using an amplitude modulated signal and added the results as Supplementary Note 7 and Supplementary Figure 9. This demonstrates that the CCDD approach can detect signals with multiple frequency components. We have added a reference to this measurement in the main text at P. 14, L 272: “Notably, a single measurement can resolve multiple frequencies (see Supplementary Note 7 and Supplementary Figure 9), demonstrating that CCDD could be applied to the detection of more complex signals that use carrier waves with frequency or amplitude modulation, for example.”

Additionally, we have added a paragraph to the discussion where we consider the potential impact and mitigation of environmental noise associated with condensed matter systems. This can be found at P. 17, L. 340 in the revised manuscript: “For example, spin waves resonant with our device could be optically excited [34, 35] at one end of the CPW. As these spin waves propagate with $\leq 100 \mu\text{m}$ wavelengths [36], the phase dependent sensor response could be used with time correlated laser scanning confocal microscopy to image the spin waveform along the length of the CPW, potentially providing new insight into spin wave dispersion relations [34, 37]. Operating above saturation magnetisation would protect the defect ensemble from spin mixing produced by ferromagnetic in-plane fields. Although thermal and incoherently driven magnon modes are known to degrade T_1 times, which would limit the sensitivity of the scheme, this can be partially offset by displacing the ensemble by a small distance from the ferromagnetic surface [38]. Additionally, any associated increase in inhomogeneous noise can be decoupled with the CCDD drive, provided that the modulation amplitude is larger than the inhomogeneous linewidth [23].”

3. There are two Methods sections—one on page 4 and another on page 16. Please consolidate these sections for clarity and coherence.

(R1.3) The first methods section has been removed, and the *Experimental Setup* and *Phase detection using a double microwave drive* sub-sections are now included in the *Results* section, which is consistent with Nature Communications formatting.

Reviewer #2 (Remarks to the Author):

The manuscript by Patrickson et. al extends their previous work (npj Quantum Information 10, 5 (2024)), which demonstrated high-frequency (up to GHz) RF/MW field sensing using VB- defects in hBN and a continuous concatenated dynamic decoupling (CCDD) scheme.

The additional feature shown here involves the sensitivity of the protocol to the signal phase, leading to the implementation of a heterodyne-like scheme for RF/MW detection with high spectral resolution. The manuscript is interesting, showcasing high-quality experiments and data.

However, I found some critical points that I would like to highlight:

Major:

- I am quite confused about how the claimed coherence protection contributes to the sensitivity of the method. For example, on line 59: “... we present a continuous drive sensing protocol that can detect the phase, frequency and amplitude of an AC magnetic field, whilst also achieving coherence times approaching $T_2 \approx 1/2 T_1$ ”.

However, they also mention that to achieve phase sensitivity, they have to compromise (at least partially) on coherence protection. Line 118: “To sense the signal phase, we apply the second drive in phase with the first drive, perpendicular to the spin vector, This reduces the protection provided by the microwave drive [30]” .

The authors should clarify this point, as it seems to contradict their earlier claims. Is coherence protection crucial for detection or not? To what extent? Additionally, what are the relaxation time T_1 and coherence times T_2 of the system in their experiments? What is the real meaning of the time constant T_{Rabi} given in the supplementary information, and is this time constant relevant for the CCDD experiment? How?

(R2.1) The wording “*whilst also achieving*” has now been changed to “*using*”. The signal field results in an additional angular velocity acting on the spin vector, and the coherence time limits the net rotation angle, and hence the change in PL signal. For phase-sensitive detection of the signal field, the drive phase, θ_m must be set to $\frac{\pi}{2}$. In the limit of low signal ($g_x \rightarrow 0$), this results in a coherence time of $\sim 0.7 \mu\text{s}$, compared to $\sim 7.4 \mu\text{s}$ for $\theta_m = 0$, where the PL is insensitive to the phase of the signal field, see SI Fig. 1c. When a signal is applied, this also helps to protect the spin coherence, resulting in a coherence time, and sensitivity that improves with signal field strength, see SI Fig. 1(f), and Fig. 3d. The $T_1 = 15 \mu\text{s}$, see SI Fig. 1b, $T_{echo} \approx 100 \text{ ns}$, see ref. [18], and $T_{Rabi}(\Omega = 100 \text{ MHz}) = 36 \text{ ns}$, see SI Fig. 1(a).

The spin has a noise spectrum that describes the coupling to the environment. The time-constant of the coherence loss depends on the applied drive, since the rotation of the spin provides a noise filter. All time constants are defined with respect to the measurement, for example $T_{Rabi}(\Omega)$ is the time constant of a single drive Rabi oscillation of Rabi frequency Ω . For phase sensitive detection, the relevant time constant is $T_{CCDD}(\theta_m = \frac{\pi}{2}, \Omega, \epsilon_m, g_x)$. In the limit, $\epsilon_m \rightarrow 0, g_x \rightarrow 0$, $T_{CCDD}(\theta_m, \Omega, \epsilon_m = 0, g_x = 0) = T_{Rabi}(\Omega)$, which serves as a benchmark for the benefit of the phase-modulation.

To clarify these points, we have made the following changes to the manuscript:

1. We have removed the statement “*This reduces the protection provided by the microwave drive [30]. Instead, coherence stabilization is now provided by the signal field (Fig. 2(a))*” (previously at line 120-122), because the focus here is on the phase detection. It is also not the case that coherence protection is provided solely by the signal.
2. We have removed “*Note that without a signal (turquoise) the ensemble decoheres within ~ 1000 ns due to unprotected fluctuations in the CCDD drive amplitude Ω [30]. However, in our scheme this is corrected by the signal field, which behaves as an additional decoupling drive. This improves the devices sensitivity by extending spin coherence.*”, (previously at line 171-175), so that the initial discussion of Fig. 2 is focused on switching between phase sensitive and phase insensitive detection.
3. We have added a paragraph after this, starting at P. 11, L. 195, which focuses on the influence of the CCDD parameters and signal amplitude on the coherence time: “**The phase and amplitude sensitivity are expected to scale with the coherence time $T_{CCDD}(\Omega, \theta_m, \epsilon_m, g_x)$ [4]. We therefore quantify the expected sensing performance by measuring the spin coherence under different operating conditions. The upper limit is governed by $T_1 \approx 15$ μ s in this sample, whilst the lower limit is determined by the Rabi time constant $T_{Rabi}(\Omega = 100$ MHz) = $T_{CCDD}(\Omega = 100$ MHz, $\epsilon_m = 0$) ≈ 36 ns, (see Supplementary Note 1 and Supplementary Figure 1). By comparison, with no signal the CCDD drive achieves a coherence time of $T_{CCDD}(\Omega = 100$ MHz, $\theta_m = \frac{\pi}{2}$, $\epsilon_m = 10$ MHz, $g_x = 0$) ≈ 700 ns (Fig. 2(a)) and can be further improved by optimizing ϵ_m (Supplementary Figure 1(e)). When a signal is applied, the coherence time is found to improve, as the signal acts as an additional decoupling drive [4] and reaches $T_{CCDD}(\Omega = 100$ MHz, $\theta_m = \frac{\pi}{2}$, $\epsilon_m = 10$ MHz, $g_x = 400$ kHz) ≈ 4 μ s (Supplementary Figure 1(f)). This protection is not as effective as when $\theta_m = 0$, where we find $T_{CCDD}(\Omega = 100$ MHz, $\theta_m = 0$, $\epsilon_m = 10$ MHz, $g_x = 0$) ≈ 7.5 μ s, but $\theta_m = \pi/2$ is necessary for phase sensitive detection, and improves the sensitivity by $\sim 30\times$ in the limit of low signal amplitude and $>100\times$ for optimum signal amplitude.”**
4. The relevant data supporting the coherence times above has been added as Supplementary Note. 1 and Supplementary Figure 1.

- Line 121: “Instead, coherence stabilization is now provided by the signal field (Fig. 2(a)).” It would be appropriate to cite the work of Stark et al. (Nature Communications 1105, 2017), as they addressed a similar concept.

The limits of this approach remain unclear to me. Specifically, how strong does the signal field need to be to function as an auxiliary protective field? Can this method still be effective in realistic sensing scenarios with very weak AC fields?

(R2.2) We now also cite Stark et al [4] at this point in the manuscript. As shown in new SI Fig. 1f, a signal strength of about $g_x = 100$ kHz (~ 3.6 μ T) doubles the coherence time $T_{CCDD}(\Omega, \theta_m = \pi/2, \epsilon_m = 10$ MHz), and hence acts like an auxiliary protective field for similar signal amplitudes. This leads to an improvement in phase sensitivity with signal amplitude, see Fig 3d. In the limit of vanishing signal, we measure an optimum value of $T_{CCDD}(\Omega = 100$ MHz, $\theta_m = \pi/2, \epsilon_m = 13$ MHz) ≈ 1200 ns (Supplementary Figure 1(e)). In the small signal limit, this coherence time, along with the intensity of the PL signal and the integration time will determine the minimum signal amplitude that can be detected. The smallest field we have so far measured is ~ 2 μ T.

To highlight that the phase sensitivity depends on the signal amplitude we have added the following statement to the discussion of Fig. 3(d) at P. 12 L. 234: “The phase sensitivity η_ϕ is characterised as a function of signal amplitude g_x , and plotted in Fig. 3(d). As before, the sensitivity is calculated using the contrast ΔC_0 shown in the inset to Fig. 3(d). The sensitivity improves with signal amplitude due to a combination of the oscillatory response of the contrast, and a coherence time that improves with signal amplitude. A minimum in the phase sensitivity of $\eta_\phi = 0.076 \text{ rads}/\sqrt{\text{Hz}}$ is reached for signal amplitudes of $g_x \approx 35 \text{ } \mu\text{T}$. For a given signal amplitude, the sensitivity can be optimised by selecting $\$T_{\text{MW}}\$$ to reduce the total measurement time, t_m (see Supplementary Note 5). Note that the signal amplitude can also be measured independently of signal phase by choosing $\theta_m = 0$ [8].

To further address this question, we have characterised the heterodyne SNR as a function of signal amplitude, see SI Fig. 8, and added supplementary note 6.

- Fig. 1 and description of spin dynamics: I acknowledge the authors' efforts to describe the spin dynamics as thoroughly as possible. However, to fully understand the experiments and the physics involved, I had to refer back to the literature, particularly the previous npj Quantum Information article by the same authors. The manuscript does not seem self-sufficient in this respect. At a minimum, the basic principles should be included so that readers can grasp the physics underlying the experiment.

(R2.3) – To clarify the underlying dynamics we have added additional steps in the mathematical description provided in P. 7 L. 135 –143.

- While I appreciate the organization of Figure 1, the authors should label the axes in the Bloch spheres shown in Figure 1(d). It would also be helpful to divide Figure 1(d) into labeled sub-figures, and refer to these sub-figures accordingly throughout the main text.

(R2.4) – The figure has been amended as suggested. The axes have been labelled on only the left-hand Bloch sphere to maintain the overall clarity of the figure. The Z-projection spin readout plots have been labelled as sub-panels (e) and (f).

- Line 56: “Alternatively, single continuous drives have been used to replicate the phase response, [22, 26 (<https://journals.aps.org/pr/abstract/10.1103/PhysRevA.104.L020602>)], but the sensitivity of these schemes is limited by coherence times $\ll T_2$.

Again, on Line 289: “A continuous scheme is also presented in [22-<https://www.nature.com/articles/s41467-021-22714-y>], however, the single drive does not extend the coherence time, limiting the sensitivity.”

Are the authors sure about this? Aren't these methods rather limited by $T_{1\rho}$, which is typically much longer than T_2 ? Isn't that an extension of the coherence? I would suggest double checking this point.

(R2.5) – The reviewer is right that there is an extension of the coherence time in these schemes. In ref. [26], the phase of the signal field determines whether it acts on the spin vector as a spin lock, or Rabi drive. In both cases this is an extension of the coherence, with $T_{1,\rho} \gg T_{Rabi} \gg T_2^*$, but coherence protection is solely due to the signal. In the limit of vanishing signal amplitude, the scheme in ref. [26] is therefore limited by T_2^* . In ref. [22], there are two schemes. In the absence of signal, the “continuous Floquet” scheme uses a single drive and the sensitivity is limited by $T_{Rabi} \gg T_2^*$, whereas the “pulsed-mollow” scheme is limited by $T_{1,\rho}$, but the measured coherence time is not specified. For comparison, and as discussed above, we find $T_{Rabi}(\Omega = 100 \text{ MHz}) = 36 \text{ ns} \ll T_{CCDD}(\Omega, \theta_m = \frac{\pi}{2}, \epsilon_m = 13 \text{ MHz}, g_x = 0) \approx 1200 \text{ ns} \approx 0.08 T_1$. Note that in the revised manuscript ref. [26] is now ref. [28], and ref. [22] is now ref. [13]. We have amended the discussion to include a direct comparison to these papers at P. 16, L. 309: “To compare our CCDD-based scheme with previous demonstrations of GHz heterodyne sensing [13, 28], we consider the coherence time in the limit of low signal amplitude. In Staudenmaier et al [28], this is T_2^* . In Meinel et al. [13], two schemes are presented: a continuous scheme, which is limited by $T_{Rabi} \gg T_2^*$, and a pulsed scheme, which is ultimately limited by T_1 but depends on the number of pulses used in the dynamical decoupling sequence.”

- Line 280 - In my opinion this statement needs to be double-checked: “The well-established CASR protocol[19] was recently demonstrated using a VB ensemble in hBN for the first time[38], where the authors achieved a frequency resolution of 0.9 Hz for an 18 MHz signal sampled for 2000 s, compared with 0.118 Hz resolution for a 10 s integration in this work.”

It's easy to verify that the authors of [38] sampled their signal for 2 seconds, not 2000 seconds (as shown in one of the main figures in their paper, Rizzato et al., Nat. Commun. 14, 5089 (2023)).

The reviewer is correct. In ref. [38] the signal is sampled for 2 s, but in the main paper this is averaged over 1000 repetitions to improve the signal to noise ratio (SNR). The sampling time determines the resolution, and the averaging improves the SNR. We have therefore provided a more detailed comparison in our revised manuscript, based on the sampling time of 2 s, at P. 15, L. 298 (note that ref. [38] is now ref. [9] in the revised manuscript): “Finally, by embedding the sensing protocol in a quantum heterodyne measurement, we have measured a $\sim 2.3 \text{ GHz}$ signal with a resolution of $\sim 0.1 \text{ Hz}$ and SNR of 235, for total measurement time of 10 s. In previous work with V_B^- ensembles in hBN [9], the coherently averaged synchronized readout (CASR) protocol [9] was used to

detect an 18 MHz signal with resolution of 0.9 Hz and SNR of 8.5 for a 2 s sampling time. In both cases, the SNR is limited by the small sample volume.”

Minor:

Supplementary Note 1 was never mentioned throughout the text.

This has been corrected.

In the Supplementary Information, the title of the manuscript is missing, as well as the headings of Supplementary Notes 1, 2, and 3, etc...

Supplementary Information is now titled, and all Supplementary Notes have headings.

Methods: It is good practice to state not only what kind of instrumentation was used, but also what specific equipment was used, so that one can, in principle, try to reproduce the experiments using the same machines.

These details have been added to the Methods section.

Line 330: Double parentheses in reference 22.

Extra parentheses removed

Reviewer #3 (Remarks to the Author):

In their manuscript, Patrickson et al. report on a protocol for the detection of microwave (GHz) signals using defect spin qubits (ensemble of boron vacancies in hBN) as quantum sensors. The protocol is based on a continuous concatenated dynamical decoupling (CCDD). In addition to sensing the microwave amplitude, by selecting a proper phase of the second drive, the sensitivity to the signal phase can be attained. The protocol can also be combined with the quantum heterodyne scheme, and in this case provides a frequency resolution of 0.118 Hz in a 10-sec measurement.

The detection of both amplitudes and phases of high-frequency signals with high sensitivities is of importance in expanding the application range of quantum sensing. The manuscript is overall well-written, and the experimental data is rich and high quality. What bothers me however is the way in which the introduction is written. In my view, it is not very readable as well as failing to position the present work in a proper framework. So, the revision is necessary to reach my acceptance/rejection decision. See my comments below.

In the first paragraph, the authors begin with the choice of the materials, listing up dense NV centers in bulk, NV centers in nanodiamonds, B vacancies and "single carbon-related defects of ambiguous structure" in hBN. These are quite remote from the main topic of the present work, CCDD-based high-frequency sensing. I understand that the coherence of a quantum sensor is a limiting factor to achieve high sensitivities, and the CCDD-based protocol is advantageous when the intrinsic coherence time of the sensor is short, such as in the case of B-vacancy in hBN used in the present work. But this is a bonus; what is more important is that the protocol itself is independent of the choice of the materials (so has wide applicability). I do not see a good reason to start the introduction this way.

(R3.1) We appreciate the different perspective and have rewritten the introduction along the lines suggested, P. 2 L. 27 – 81. The title has also been changed to "Microwave quantum heterodyne sensing using a continuous concatenated dynamical decoupling protocol", to move the focus from the material system to the protocol.

The second paragraph outlines the existing sensing protocols, but it looks to me that the authors are trying to unduly undermine them. For instance, quoting Refs. [23] and [24], the authors state "pulsed methods are sub-optimal, as the sensitivity suffers from pulse-area errors." It is true that one should pay careful attention to the consequence of finite pulse lengths, but the issue raised in Ref. [24] can be trivially solved by including the pulse lengths in the total sensing time (which is almost always done). To highlight the effect, the examples in Ref. [24] is taken from the cases when the pulse durations are very long (~ 1 us), not the condition where many practical sensing experiments are carried out. The detuning effect discussed in Ref. [23] may be potentially more serious, but the effect is pronounced when the total number of pulses is large and XY8 is shown at least to suppress the spectral splitting. On the other hand, the quantum heterodyne protocol utilizes a short pulse sequence repeated with a constant interval. I am so far not aware of the cases when the quantum heterodyne protocols have practically suffered the effect of finite pulse lengths. If known, please cite appropriate references. Also, in this paragraph, the authors mention that the CCDD-based protocol achieves the effective coherence limit of $T_2 \sim (1/2)T_1$. Please provide (not necessarily in the introduction) the concrete values of T_1 and T_2 demonstrated by the current CCDD-based sensing experiments.

(R3.2) We did not intend to undermine pulsed techniques, but rather to highlight some potential advantages of our CCDD method. In re-writing the introduction we have removed the generalised comparison between pulsed and continuous dynamical decoupling. Instead, we have added a section to the discussion, where we have focused on comparing our CCDD scheme to pulsed dynamical decoupling, specifically for systems with large inhomogeneous broadening, such as V_B^- in hBN. This can be found at P. 16 L. 309 – 333.

We have added a comprehensive overview of the coherence times as Supplementary Note 1 and Supplementary Figure 1, which includes additional measurements of the coherence time under CCDD drive (see also R2.1). The measurements show that $T_1 = 15 \mu\text{s}$ and the time constant of a single drive Rabi oscillation with Rabi frequency $\Omega = 100 \text{ MHz}$ is $T_{Rabi}(\Omega = 100 \text{ MHz}) = 36 \text{ ns}$. In the case of phase sensitive detection, when $\theta_m = \frac{\pi}{2}$, the maximum measured coherence time is $T_{CCDD}(\Omega = 100 \text{ MHz}, \theta_m = \frac{\pi}{2}, \epsilon_m = 10 \text{ MHz}, g_x = 0) \approx 1200 \text{ ns}$, in the absence of a signal. This increases to $T_{CCDD}(\Omega = 100 \text{ MHz}, \theta_m = \frac{\pi}{2}, \epsilon_m = 10 \text{ MHz}, g_x = 400 \text{ kHz}) \approx 4 \mu\text{s}$ when a signal is applied. To clarify this in the manuscript, we have added the following statement in the discussion at P. 15 L. 294: “For the phase sensitive detection mode, the coherence time is improved from $T_{Rabi} \approx 36 \text{ ns}$ to $T_{CCDD} \approx 1.2 \mu\text{s}$ with no signal, and to $T_{CCDD} \gtrsim 4 \mu\text{s} \approx 0.25T_1$ for an optimum signal amplitude.”

We have also added a detailed appraisal of the coherence times at P. 11, L. 195: “The phase and amplitude sensitivity are expected to scale with the coherence time $T_{CCDD}(\Omega, \theta_m, \epsilon_m, g_x)$ [4]. We therefore quantify the expected sensing performance by measuring the spin coherence under different operating conditions. The upper limit is governed by $T_1 \approx 15 \mu\text{s}$ in this sample, whilst the lower limit is determined by the Rabi time constant $T_{Rabi}(\Omega = 100 \text{ MHz}) = T_{CCDD}(\Omega = 100 \text{ MHz}, \epsilon_m = 0) \approx 36 \text{ ns}$, (see Supplementary Note 1 and Supplementary Figure 1). By comparison, with no signal the CCDD drive achieves a coherence time of $T_{CCDD}(\Omega = 100 \text{ MHz}, \theta_m = \frac{\pi}{2}, \epsilon_m = 10 \text{ MHz}, g_x = 0) \approx 700 \text{ ns}$ (Fig. 2(a)) and can be further improved by optimizing ϵ_m (Supplementary Figure 1(e)). When a signal is applied, the coherence time is found to improve, as the signal acts as an additional decoupling drive [4] and reaches $T_{CCDD}(\Omega = 100 \text{ MHz}, \theta_m = \frac{\pi}{2}, \epsilon_m = 10 \text{ MHz}, g_x = 400 \text{ kHz}) \approx 4 \mu\text{s}$ (Supplementary Figure 1(f)). This protection is not as effective as when $\theta_m = 0$, where we find $T_{CCDD}(\Omega = 100 \text{ MHz}, \theta_m = 0, \epsilon_m = 10 \text{ MHz}, g_x = 0) \approx 7.5 \mu\text{s}$, but $\theta_m = \pi/2$ is necessary for phase sensitive detection, and improves the sensitivity by $\sim 30\times$ in the limit of low signal amplitude and $>100\times$ for optimum signal amplitude.”

In the third paragraph, the authors introduce their protocol, starting with "In this work, ..." However, the key idea comes from CCDD, which has been extensively worked out in the past, in particular in Ref. [27] (Cai et al 2012). Of course, Ref. [27] (Cai et al 2012) is cited, but why wasn't CCDD discussed before "In this work"? The introduction must clearly separate the past achievements from the present work. In fact, I feel at the conceptual level the novelty of the present protocol is not so high. In addition to Ref. [27] (Cai et al 2012), CCDD-based sensing in the GHz range has been demonstrated in Ref. [29] (Stark et al 2017). In comparison to Ref. [29] (Stark et al 2017), the authors pitch that their protocol can sense the microwave phase as well. But this requires the signal being highly coherent. Rather, Ref. [29] (Stark et al 2017) points out that CCDD-based sensing can be done even when the

driving and signal fields are not phase-matched so that the sensing is generic and widely applicable. These different viewpoints should be more clearly stated. The present work also has a significant overlap with the authors' own work of Ref. [34]. It is heavily cited in the technical part, but not in the introduction. Again, the difference between Ref. [34] and the present work should be clarified in the introduction. What is the conceptual advance, other than moving to the higher frequencies?

(R3.3) Our work is influenced by refs. 27 and 29. However, in refs. 27,29,34, the measurement protocols presented measure the amplitude of the ac-signal. Since, these protocols are insensitive to phase, it does not require a coherent signal, and for many applications this is desirable. The measurement protocol presented here is sensitive to a quadrature of the ac-field, and by definition requires a coherent signal. For a different set of applications, where information on the signal phase with respect to reference field is desired, a coherent signal will be a feature, and it is the phase sensitivity which allows the CCDD scheme to be used in high resolution heterodyne sensing. Phase detection is the conceptual difference between the current work and refs. [27,29,34]. To highlight this point we have emphasised the novelty of this work (3rd paragraph) in comparison to prior reports of CCDD sensing (summarised in 2nd paragraph). Note that refs. [27,29,34] are now refs. [15,4,8] in the revised manuscript, respectively.

Minor errors I have noticed are listed below.

All minor errors have been addressed accordingly.

(1) There are multiple comments like "see Supplementary Note #" in the main text, but the sections of Supplementary Information (SI) are not numbered. Please add the section numbers.

(2) After the equations, "where" should start without spaces. (And in this case a comma should be added after the equations.)

(3) In some situations, the units for the amplitudes and phase sensitivities are given as " $\text{uT} \sqrt{\text{Hz}}$ " and " $\text{rads} \sqrt{\text{Hz}}$," respectively. Please check and correct them.

(4) In Line 57, "... the phase response, [22, 26], but..." should read "... the phase response [22, 26], but..."

(5) In Line 83, "A 488nm laser" should read "A 488-nm laser."

(6) In Line 84, "NA=0.55" should read "NA = 0.55."

(7) In Line 88, "room temperature in air. (see...)." should read " room temperature in air (see...)."

(8) In Line 99, ω_0 as calculated from the given D, E, ω_{-1} does not give 2.32 GHz. Note that ω_{-1} is given as negative.

(9) In Line 220, " $\eta_{\phi} = 0.076 \text{rads} / \sqrt{\text{Hz}}$ " should read " $\eta_{\phi} = 0.076 \text{rads} / \sqrt{\text{Hz}}$."

(10) In Line 254, "SNR=235" should read "SNR = 235."

(11) In Line 260, "950 ns to 10s of ns" should read "950 ns to tens of ns," to avoid confusion.

(12) In Line 267, the authors write "we have demonstrated a phase modulated CCDD sensing protocol...". As far as I can see, the authors controlled the drive phase θ_m , but this is hardly a phase modulation, which would usually mean the phase being time-dependent.

(R3.12) Here, phase modulation refers to the general CCDD control field described by H_C in Eq. (1):

$$H_C = \Omega \cos(\omega_0 - \frac{2\varepsilon_m}{\Omega} \sin(\omega_m t - \theta_m)) \sigma_x,$$

where the concatenated time-dependent term with frequency ω_m is considered to be a phase modulation of the main drive.

(13) In Line 315, "a 488 nm diode laser" should read "a 488-nm diode laser."

(14) In Line 316, "N.A.=0.55" should read "N.A. = 0.55." Also, the use of both "N.A." and "NA" (Line 84) should be avoided.

(15) In Line 330, "constraints [[22]]" should read "constraints [22]."

(16) In Line 333, " $t_{\text{gate}} = 350\text{ns}$ " should read " $t_{\text{gate}} = 350 \text{ ns}$."

(17) In the caption of Fig. 1 of SI, it will be better to include the coefficient in front of $\sin(\omega t)$. " $T_{\text{Rabi}} = 36\text{ns}$ " should read " $T_{\text{Rabi}} = 36 \text{ ns}$."

(18) In the caption of Fig. 3 of SI, " $T_{\text{M}}\text{W}$ " should read " T_{MW} ."

(19) In Line 21 of SI, "This illustrates our protocols dependence on..." may mean "This illustrates our protocol's dependence on..." See also Line 75 of SI.

(20) In Line 50 of SI, "it's direction of propagation" should read "its direction of propagation."

(21) In Line 113 of SI, "125ns" should read "125 ns." (And the unit should not be italicized.)

Response to Reviewer Comments

We thank all three reviewers for their thorough reviews on our manuscript, for their positive comments and constructive criticism. In the following we provide a point-by-point response, detailing where we have amended the manuscript.

Reviewer #1 (Remarks to the Author):

After reviewing the revised manuscript, I confirm that all the points I raised in the previous round have been addressed. The authors have clarified the issues and adjusted the manuscript accordingly. I reiterate my recommendation for publication in Nature Communications.

We thank the reviewer for their comments and time taken to review the manuscript.

Reviewer #2 (Remarks to the Author):

While I find the manuscript somewhat incremental compared to the authors' previous works, the results presented are solid overall. In this revised version, the authors have addressed my suggestions thoroughly. Combined with the integrated feedback from other reviewers, these changes make the manuscript substantially more comprehensive and complete, while appreciably enhancing its clarity and strengthening the overall conclusions.

I do have one remaining request. Throughout the manuscript, the authors have extensively demonstrated the method's performance for sensing signals around 2 GHz. Moreover, in the conclusion, they primarily highlight the detection of spin waves in 2D materials as a specific application—one that benefits from sensing in the GHz range. However, since the manuscript (beginning with the title) appears to have shifted toward a more general development of a sensitive and versatile sensing protocol, I would appreciate further comments on how the technique performs in lower frequency regimes (e.g., 10–150 MHz). While the authors note that their protocol is capable of detecting signals in this window, they do not elaborate on its actual performance there. This frequency range would be highly relevant for another important application, such as magnetic resonance sensing. Such a discussion would be of considerable interest to a large segment of the quantum sensing community.

After addressing this point, I recommend acceptance of the manuscript.

We thank the reviewer for their comments and agree that the manuscript would benefit from a discussion of these lower frequency resonances and their potential application to magnetic resonance schemes. We have added a paragraph to the discussion on P.17 L. 335 – 346 in the revised manuscript.

“We note that the protocol also supports lower frequency resonances at ϵ_m and $\Omega \pm \epsilon_m$ [8], which could be relevant to nanoscale NMR. ϵ_m and Ω are responsible for coherence protection, and so the choice of spin system limits the frequency range of these resonances. Here we set $\epsilon_m = 10$ MHz, which we expect to be the minimum required to protect a V_B^- ensemble in an hBN sample of natural composition [18]. For NV centres in diamond, the minimum can be set as low as $\epsilon_m \approx 100$'s kHz [4]. We expect these resonances to have similar amplitude and phase sensitivity to the GHz sensor resonances presented here, although we note that the ϵ_m resonance attenuates signal amplitudes by

$g/4$ [8], rather than $g/2$ (Fig. 2 (c)). The high Rabi frequencies achieved in this work could enable nanoscale NMR at large magnetic fields <1 T, accessing NMR frequencies in the 10's MHz, in comparison to existing pulsed schemes which are limited to NMR frequencies below a few MHz [10–12].”.

Reviewer #3 (Remarks to the Author):

I am satisfied with the revisions made by the authors. The revised introduction provides a concise summary of the existing quantum sensing protocols whilst the advantage of the present protocol is more clearly described. Supplementary Note 1 and Supplementary Fig. 1 is also a concise summary of coherence protection provided by this protocol. In my view, a separation of these two (related but different) aspects made the manuscript more readable. I recommend the present manuscript for publication in Nature Communications.

We thank the reviewer for their complimentary feedback and for the time taken to review the manuscript.